# Deprivation Index and Lifestyle: Baseline Cross-Sectional Analysis of the PREDIMED-Plus Catalonia Study

**DOI:** 10.3390/nu13103408

**Published:** 2021-09-27

**Authors:** Josep Basora, Felipe Villalobos, Meritxell Pallejà-Millán, Nancy Babio, Albert Goday, María Dolores Zomeño, Xavier Pintó, Emilio Sacanella, Jordi Salas-Salvadó

**Affiliations:** 1Fundació Institut Universitari per a la Recerca a l’Atenció Primària de Salut Jordi Gol i Gurina (IDIAPJGol), 08007 Barcelona, Spain; 2Unitat de Nutrició Humana, Departament de Bioquímica i Biotecnologia, Universitat Rovira i Virgili, 43201 Reus, Spain; jordi.salas@urv.cat; 3Consorcio CIBER, M.P. Fisiopatología de la Obesidad y Nutrición (CIBERObn), Instituto de Salud Carlos III (ISCIII), 28029 Madrid, Spain; agoday@parcdesalutmar.cat (A.G.); mzomeno@imim.es (M.D.Z.); xpinto@bellvitgehospital.cat (X.P.); esacane@clinic.cat (E.S.); 4Unitat de Suport a la Recerca Tarragona-Reus, Fundació Institut Universitari per a la Recerca a l’Atenció Primària de Salut Jordi Gol i Gurina (IDIAPJGol), 43202 Reus, Spain; fvillalobos@idiapjgol.info (F.V.); mpalleja@idiapjgol.info (M.P.-M.); 5Institut d’Investigació Sanitària Pere i Virgili (IISPV), Hospital Universitari San Joan de Reus, 43204 Reus, Spain; 6Servei d’Endocrinologia, Hospital del Mar, 08003 Barcelona, Spain; 7Cardiovascular Risk and Nutrition Research Group, Hospital del Mar Medical Research Institute (IMIM), 08003 Barcelona, Spain; 8Departament de Medicina, Univerisitat Autònoma de Barcelona, 08193 Barcelona, Spain; 9Nutrició Humana i Dietètica, Facultat de Ciències de la Salut Blanquerna-Universitat Ramon Llull, 08022 Barcelona, Spain; 10Lipid Unit, Department of Internal Medicine, Bellvitge Biomedical Research Institute (IDIBELL)-Hospital Universitari de Bellvitge, 08908 L’Hospitalet de Llobregat, Spain; 11Department of Internal Medicine, Hospital Clínic, 08036 Barcelona, Spain; 12School of Medicine, University of Barcelona, 08036 Barcelona, Spain

**Keywords:** deprivation index, lifestyle, diet, physical activity

## Abstract

This baseline cross-sectional analysis from data acquired in a sub-sample of the PREDIMED-Plus study participants aimed to evaluate the relation between the Composite Socioeconomic Index (CSI) and lifestyle (diet and physical activity). This study involved 1512 participants (759 (52.2%) women) between 55 and 80 years with overweight/obesity and metabolic syndrome assigned to 137 primary healthcare centers in Catalonia, Spain. CSI and lifestyle (diet and physical activity) were assessed. Multiple linear regression or multinomial regression were applied to the data. Cluster analysis was performed to identify dietary patterns. The multiple linear regression model showed that a high deprivation index was related to a higher consumption of refined cereals (11.98 g/d, *p*-value = 0.001) and potatoes (6.68 g/d, *p*-value = 0.001), and to a lower consumption of fruits (−17.52 g/d, *p*-value = 0.036), and coffee and tea (−8.03 g/d, *p*-value = 0.013). Two a posteriori dietary patterns were identified by cluster analysis and labeled as “healthy” and “unhealthy”. In addition, the multinomial regression model showed that a high deprivation index was related to an unhealthy dietary pattern and low physical activity (OR 1.42 [95% CI 1.06–1.89]; *p*-value < 0.05). In conclusion, a high deprivation index was related to an unhealthy lifestyle (diet and physical activity) in PREDIMED-Plus study participants.

## 1. Introduction

Lifestyles—diet and physical activity—impact on a wider range of health outcomes. A recent review concluded that more than 11 million deaths were attributable to dietary risk factors in 2017 [1]. In addition, physical activity is also a major and globally relevant determinant of health, likewise a lack of physical activity is considered a key risk factor for the development of chronic diseases and premature mortality [2].

Socioeconomic status (SES) is an important determinant of health. The relationship that exist between the socioeconomic status (SES) and several health outcomes is well known. Individuals with low SES are at higher risk of chronic diseases (physical and mental health diseases) and have lower life expectancy compared to those with high SES [3].

A relationship between SES and lifestyle have been largely documented. An individual can choose a healthy or an unhealthy lifestyle, but this choice is determined by their SES and other social determinants such as age, sex, or civil status, leading to poor or good health [4,5]. Therefore, lifestyle could mediate the relationship between SES and health [6].

Previous studies have reported an association between SES and other social determinants, and lifestyles. Regarding diet, for example, an association between a high level of education and higher consumption of fruits and vegetables has been reported [7]. Men consumed more dairy products, olives, nuts and seeds, red meat and processed food, sweets, eggs, alcohol and fast food compared to women, while women consumed more fruits; and men with low SES have a higher consumption of alcoholic beverages, compared to women [8]. On the other hand, studies that have evaluated the relationship between SES and other social determinants, and dietary patterns have observed inconsistent results [9,10,11,12,13,14]. Some studies have reported higher adherence to a healthy dietary pattern in older individuals with high SES [9,10,11,12,13,14]. On the contrary, studies have frequently reported an increased risk of adherence to an unhealthy dietary pattern in women, married and with family, non-workers, and with high education level [15,16,17]. In relation to physical activity, low levels have been observed with more prevalence in older individuals, men, married, and with low educational level and SES [18]. In addition, physical inactivity was observed to be more frequent in individuals living in neighborhoods with low SES and deprivation [19,20].

In studies of health inequalities, SES is most commonly operationalized as either education, social class or income, and often without providing a rationale for the choice of indicator. Education, social class, or income can have overlapping properties in relation to health [21]. The deprivation indices are instruments used to measure health inequalities at a population level. All of them are constructed based on different socioeconomic or demographic characteristics and are used to quantify the socioeconomic variation in health outcomes. They measure socioeconomic vulnerability and some of them make it possible to prioritize services and corrective actions. They started to be used in the 1980s in the United Kingdom (UK). The most well-known are the Townsend [22], and Carstairs and Morris indices [23], however, in the last few years other indices have been developed and validated such us the SIMD16 index in Scotland, the ID2007 in UK, the NZDEP2018 in New Zealand, the MEDEAS index in Spain [24], and the Composite Socioeconomic Index (CSI) in Catalonia, Spain [25].

The CSI is a deprivation index for the assignation of the budgets of the primary healthcare areas in Catalonia (Spain) valid both in urban and rural areas. The variables used to construct the index allow frequent updating and are representative at the territorial level of primary healthcare: exemption from pharmaceutical co-payment, income below €18,000/year, income higher than €100,000/year, manual occupations, low level of education, mortality before the age of 75 and potentially avoidable hospitalizations [25]. The components of the CSI have demonstrated an association between low socioeconomic level and high morbidity rates, high use of primary healthcare services, hospital and psychiatric care, as well as a greater use of drugs, especially for mental health problems [26]. However, there is no scientific evidence on the relationship between CSI with the lifestyle, dietary aspects nor with physical activity.

The deprivation index is considered a good instrument for classifying the SES. The CSI is built using different socioeconomic indicators, and those have shown overlapping properties when they are used individually to measure health inequalities. Demonstrating that the CSI is related to patterns of unhealthy lifestyles, diet and physical activity is important in order to use it as an instrument to design and prioritize lifestyle interventions at the community level, especially in primary healthcare areas. In addition, it would allow us to have a broad vision of how the socioeconomic contextual aspects of geographic location impact on health related to diet and physical activity. For example, neighborhood-level characteristics, such as the availability of healthy food, and the quality of the physical environment, have been proposed as determinants of the overweight and obesity prevalence [27].

Bearing in mind the aforementioned, this study aimed to evaluate the relation between CSI and lifestyle (diet and physical activity), in a sub-sample of the PREDIMED-Plus study participants.

## 2. Materials and Methods

### 2.1. Study Design

This study is a baseline cross-sectional analysis of data acquired in a sub-sample of participants enrolled in the PREDIMED-Plus study from Catalonia Health Centers. The PREDIMED-Plus study is an ongoing, multicenter, randomized, controlled clinical trial conducted in Spain involving participants between 55 and 80 years with overweight/obesity and metabolic syndrome for primary cardiovascular prevention.

The study protocol is detailed in http://predimedplus.com/, and the description of the cohort has been published elsewhere [28]. The protocol was written in accordance with the ethical principles and good clinical practices contained in the Declaration of Helsinki. This study was registered at the International Standard Randomized Controlled Trial (ISRCT; http://www.isrctn.com/ISRCTN89898870) with number 89898870. The respective Institutional Review Board of all study centers approved the study protocol and all participants provided written informed consent.

### 2.2. Participants

For the present study, we included baseline data of participants living in Catalonia (Spain) recruited and randomized from the following centers: (a) Institut Hospital del Mar d’Investigacions Mèdiques (IMIM) in Barcelona, (b) Hospital Sant Joan-IISPV/Atenció Primària in Reus, (c) Atenció Primària Metro Sur-Departament d’Aterioesclorosi de I’Hospital de Bellvitge in Barcelona, and (d) Hospital Clinic of Barcelona. The period of recruitment was from October 2013 to December 2016. Participants did not receive any type of compensation for participating in the study. The present analysis included 1512 participants (759 women) from 137 primary healthcare areas affiliated to these centers. Participants recording extreme total energy intakes (<500 or >3500 kcal/day in women or <800 or >4000 kcal/day in men) [29] and without information on CSI were excluded (*n* = 69).

### 2.3. Variables Determined

#### 2.3.1. Socio-Demographic Variables

Participants self-reported socio-demographic data: age, sex, civil status, education level and employment status.

#### 2.3.2. Composed Socioeconomic Index

The CSI was used to determine the deprivation index of the participants [25]. All primary healthcare areas registered in Catalonia (*n* = 398) have assigned a CSI. The CSI ranges from −0.01 to 5.68, and a higher value of the CSI implies higher deprivation index. For this study, we included the primary healthcare areas registered from the PREDIMED-plus study participants, the CSI ranges from −0.004 to 4.49. We classified the participants into two categories according to the CSI assigned to their corresponding registered primary healthcare area: high deprivation index (≥2.27 points) and low deprivation index (<2.27 points).

#### 2.3.3. Anthropometric Measurements

Body weight, height and waist circumference (WC) were measured by trained staff and following the PREDIMED-Plus operations protocol. Weight and height were measured using calibrated scales with participants wearing light clothes and no shoes. BMI was calculated as body weight (kg) divided by height (meters) squared. WC was measured with anthropometric tape midway between the lowest rib and the iliac crest. 

#### 2.3.4. Dietary Intake and Adherence to the Energy-Reduced Mediterranean Diet

A trained dietician asked the participants about their frequency of consumption for a specified serving size of each 143 items food frequency questionnaire item during the preceding year in a face-to-face interview [30]. For each item, a typical portion size was included, and consumption frequencies were registered in 9 categories: never or almost never, 1–3 times/month, once per week, 2–4 times/week, 5–6 times/week, once per day, 2–3 times/day, 4–6 times/day, and >6 times/day. Reported frequencies of food consumption were converted into frequencies per day, and multiplied by the weight of the typical portion size indicated to obtain the intake in g/d. To identify dietary patterns, 143 food items from the questionnaire were categorized in 23 food groups (Appendix A).

Adherence to the energy-reduced Mediterranean Diet (er-MedDiet) was assessed by trained dieticians using a recently validated questionnaire of 17 items [31]. This questionnaire has been used in the ongoing PREDIMED-Plus study aiming to assess the effect of an er-MedDiet on cardiovascular events in people with overweight and obesity at increased risk of CVD. The er-MedDiet questionnaire includes 14 items on food consumption and three items on eating behaviors, with some of the items belonging to the MEDAS validated questionnaire measuring adherence to Mediterranean diet in the PREDIMED study [32].

#### 2.3.5. Physical Activity

Physical activity was measured using the Minnesota Questionnaire validated for the Spanish population [33,34]. Intensity (light, moderate, or vigorous), frequency (days per week) and duration of physical activity (minutes per day) were registered. The intensity and frequency of each activity was used to calculate the intensity category in terms of metabolic equivalents (METs)/min/week. These values were obtained by multiplying the average energy expenditure (3.3 MET for walking, 4.0 MET for moderate intensity, and 8.0 MET for vigorous intensity) by min/week for each physical activity category. The results of each category of activity intensity were summed to obtain the total physical activity. Based on total physical activity, participants were classified into two categories: low physical activity (≤2100 METs/min/week) and high physical activity (>2100 METs/min/week).

#### 2.3.6. Sedentary Lifestyle, Smoking Habits and Clinical Morbidities

Sedentary lifestyle was measured using the Nurses’ Health Study questionnaire validated for the Spanish population [35], consisting of a set of questions assessing the average daily time spent over the last year watching TV, sitting while using the computer, sitting during journeys, and total sitting. Answers included 12 categories ranging from never to ≥9 h/day of sitting time for the corresponding activity. A sedentary lifestyle was defined as ≥7 h/day of sitting time. Furthermore, participants reported their average daily sleeping time for both weekdays and weekends, using the non-validated open question, “How many hours do you sleep on average per day on weekdays and weekends?” Additional information related to smoking habits and clinical morbidities (presence of self-reported hypertension, dyslipidemia and type 2 diabetes mellitus) was collected.

### 2.4. Statistical Analysis

The data are presented as mean and standard deviation (SD) for continuous variables, or as a median and interquartile range [IR] for non-normally distributed data, and frequencies and percentages for categorical variables. Variables of the study were compared across different groups: CSI, food groups and lifestyles categories. We used *t*-tests or ANOVA-tests for comparisons of continuous variables among groups. The Mann–Whitney U test or the Kruskall–Wallis test was employed for the continuous variables that did not have a normal distribution according to the Kolmogorov–Smirnov test. For the pairwise comparison, corrected for multiple comparisons, the Tukey method was used when explanatory variables were normal-distributed and the Benjamini and Hochberg method otherwise. Comparisons among groups for categorical variables were performed with the χ^2^ test and Fisher test when the expected frequencies were less than five.

The calorie-adjusted nutrient intake was made to avoid bias produced by the inter-individuals’ variability of energy intake [36].

The relation between the CSI categories (low/high deprivation index) as exposure and the food consumption (food group, g/day) as outcome, was evaluated by multiple linear regression models adjusted by age (years), sex (man/woman), smoking (smoker, former smoker or never smoked), waist circumference (cm), physical activity (low/high), sedentary lifestyle (no/yes), hypertension (no/yes), dyslipidemia (no/yes) and type 2 diabetes mellitus (no/yes).

Cluster analysis, using the K-means method was performed to derive dietary patterns. The K-means method was applied based on Euclidean distances, and the data was input as z-scores. Two clusters were specified prior to analysis. Participants were divided based on the similarity of their food consumption (food groups adjusted for standardized energy).

By combining the dietary patterns created by cluster analysis (“healthy” and “unhealthy”), and the physical activity categories (low/high), four categories were created reflecting the lifestyle of the participants. In this way, each group of participants had to accomplish both conditions: be in the specified dietary pattern and specified physical activity category.

The relation between the CSI categories (low/high deprivation index) as outcome and the lifestyle (identified dietary pattern and physical activity category) as exposure, was evaluated by a multinomial regression model adjusted by age (years), sex (man/woman), smoking (smoker, former smoker or never smoked), waist circumference (cm), sedentary lifestyle (no/yes), hypertension (no/yes), dyslipidemia (no/yes), and type 2 diabetes mellitus (no/yes).

The selection of the covariates, which were included in the models, was based on the factors affecting choice of a healthy lifestyle [37], and on the inclusion criteria of the study, comorbidities could previously condition the sample for having received lifestyle interventions based on their risk factor.

Statistical significance was set at *p*-value < 0.05. Analyses were performed with the statistical software “R 4.03” for Windows.

## 3. Results

### 3.1. General Characteristics of the Participants and the Composite Socioeconomic Index (CSI) Categories

Table 1 shows the general characteristics of the participants according to the CSI categories. There were significant differences with respect to age, education level, employment status, adherence to the erMedDiet, physical activity, sedentary lifestyle and hypertension. Specifically, a higher percentage of participants with a high deprivation index compared to those with a low deprivation index, had a lower educational level and were not currently working. In addition, they had lower adherence to the erMedDiet, practiced less light and total physical activity, and a higher percentage of them had a sedentary lifestyle and hypertension.

### 3.2. Food Consumption of the Participants and the CSI Categories

Table 2 shows the food consumption of the participants according to the CSI categories. Participants with a high deprivation index compared to those with a low deprivation index had significantly lower consumption of full-fat dairy, red meat and meat products, whole grain cereals, fruits, sugar-free beverages, coffee or tea, spirits beverages and wines, and significantly higher consumption of refined cereals, potatoes, and biscuits and pastries.

Multiple linear regression models showed that being a participant with a high deprivation index was related to a higher consumption of refined cereals (*p*-value = 0.001) and potatoes (*p*-value = 0.001), and to a lower consumption of fruits (*p*-value = 0.035), and coffee and tea (*p*-value = 0.012). No significant relationships were observed between the CSI categories and the consumption of other predefined food groups (Figure 1 and Appendix A).

### 3.3. Dietary Patterns

Figure 2 shows the two identified dietary patterns by cluster analysis. Due to their characteristics and affinities, these two patterns have been labeled as “healthy” followed by 704 (46.5%) participants, and “unhealthy” followed by 808 (53.5%). The “healthy” pattern was characterized by a significantly higher consumption of low-fat dairy, white meat, fish and seafood, whole grain cereals, legumes, fruits, vegetables, nuts, olive oil and olives. The “unhealthy” pattern was characterized by a significantly higher consumption of foods rich in fat, sugar and alcohol such as full-fat dairy, red meat and meat products, refined cereals, other fat or oils different from olive oil, full-fat dairy derivatives and processed meals, potatoes, biscuits and pastries, sugar, sweets, chocolate and cocoa, cava and beers, spirits and wines (Appendix A).

### 3.4. General Characteristics of the Participants According to Predefined Lifestyle Categories

Four categories were created reflecting the lifestyle of the participants: (1) unhealthy dietary pattern and low physical activity, (2) unhealthy dietary pattern and high physical activity, (3) healthy dietary pattern and low physical activity, and (4) healthy dietary pattern and high physical activity.

Table 3 shows the general characteristics of the participants according to the predefined lifestyle categories. Significant differences were observed with respect to sex, age, employment status, BMI, waist circumference, central obesity, adherence to the erMedDiet, sedentary lifestyle, and smoking. Specifically, compared to those participants with an unhealthy dietary pattern & low physical activity, participants with a healthy dietary pattern & high physical activity were older, and were more likely to be women and retired, had lower BMI and waist circumference (men), and a lower percentage of them had central obesity; in addition, they had a higher adherence to the erMedDiet, and a lower percentage of them were sedentary and smokers.

### 3.5. Relation between CSI and Lifestyle

Table 4 shows the relationships between the CSI categories and the participant’s lifestyle. The multinomial regression model shows that being a participant with a high deprivation index was positively related to a lifestyle composed of an unhealthy dietary pattern and low physical activity (OR 1.42 [95% CI 1.06–1.89]; *p*-value < 0.05). No significant associations were observed between the CSI categories and other predefined lifestyles considered.

## 4. Discussion

In this baseline cross-sectional study conducted in PREDIMED-Plus study participants living in Catalonia, being a participant with a high deprivation index was related to a high consumption of refined cereals, potatoes, and to a lower consumption of fruits, and coffee and tea; four lifestyle categories were identified, and a high deprivation index was related with an “unhealthy” dietary pattern associated with low physical activity. These results support the limited existing evidence on the relationship between the deprivation index of a certain population area, dietary consumption and physical activity in individuals with overweight/obesity and metabolic syndrome.

Previous studies have observed that a high deprivation index is related to food consumption. A systematic review reported that individuals living in the most deprived areas had a lower consumption of fruits and vegetables [38]. Findings on relationship between measures of deprivation index and diet patterns have been inconsistent. In Australia, one study did not find any relationship between the deprivation index and the observed dietary patterns: Mediterranean, Prudent, and Western [39]; by contrast, in another study adherence to a healthy pattern (characterized by breakfast cereal, low fat milk, soy and rice milk, soup and stock, yoghurt, bananas, apples, other fruit and tea, and low consumption of pastries, potato chips, white bread, take-away foods, soft drinks, beer and wine) was inversely related to the deprivation index [10]. In a study conducted in Japan, individuals who lived in the most deprived areas had a lower score of adherence to the Japanese diet (low consumption of grains, potatoes, vegetables, fruits, mushrooms, fish, seafood; and high consumption of legumes, meat and coffee) [40].

Furthermore, our results were consistent with studies that have reported a relationship between the deprivation index and lifestyle. A study carried out in the UK observed that individuals in the highest deprivation quintile had a greater prevalence risk of adhering to an unhealthy lifestyle (low consumption of olive oil, fish, fruits and vegetables, high consumption of red and processed meat, and low physical activity) [41]. Similar observations were reported in a study in Australia, where a high deprivation index was related to an unhealthy lifestyle (less than five rations/day of fruits and vegetables, high alcohol consumption and low physical activity) [42].

The possible mechanisms related to being an individual with a high deprivation index and an “unhealthy” dietary pattern could be that individuals living in the most deprived areas suffer from so-called “food deserts” [43]. These areas are characterized by poor access to healthy and affordable food, and are characterized by social and spatial disparities in diet and diet-related health outcomes such as obesity [43]. A systematic review reported that better food access (availability, accessibility, affordability, accommodation and acceptability) is related to a healthy diet [44]. With respect to physical activity, it is recognized that green spaces accessibility may influence physical activity adherence [45]. The accessibility of greens spaces is usually better in more deprived areas but those residents have more negative perceptions (poorer perceived accessibility and poorer safety) and are less likely to use the green spaces [46].

One aspect that we can highlight from our findings is that being a woman is related with a healthy lifestyle. Previous studies support this relationship: women have reported higher adherence to a healthier diet [9,10,11,12,13,14], and higher levels of physical activity [18]. Women place greater importance on healthy eating than men, health beliefs explain a large proportion of dietary behavior, and they are more interested in and actively seek health-related information to a larger extent than men [47,48]. However other studies have reported an increased risk of adherence to an unhealthy dietary pattern in women [15,16,17]. More knowledge on gender differences in lifestyles could facilitate public health initiatives to promote healthy lifestyles in women and men.

A major strength of this study is the use of the CSI. It is a deprivation index that is valid for all basic health areas (urban and rural), it is easy to interpret, can be updated more frequently than indices constructed from census variables, and is related to the need of health service use, so it can be utilized to design and prioritize lifestyle interventions in primary healthcare, with community repercussions.

Some limitations deserve to be mentioned, such as the inherent nature of cross-sectional studies that do not allow causality to be addressed. Moreover, our results cannot be extrapolated to other populations, the findings of this study can only be applied to people with overweight and obesity at increased risk of CVD. In addition, this study could include the subjective decisions required in the use of cluster analysis, such as the number of clusters to implement the k-means algorithm, food group, and naming of dietary patterns. Lastly, a convenience sample was used, PREDIMED-Plus participants living in Catalonia.

## 5. Conclusions

This study contributes to the scarce knowledge on the relationship between the deprivation index and lifestyle in individuals with overweight/obesity and metabolic syndrome. CSI was related with lifestyle in the PREDIMED-Plus study participants living in Catalonia, Spain. Those participants with high deprivation index are at greater risk of adhering to an “unhealthy lifestyle” following an unhealthy dietary pattern and having lower physical activity. Public health policy should consider this relationship, by understanding how these factors influence lifestyle in individuals with overweight/obesity: community interventions and health policy decisions may target subsets of the population in order to promote a healthier lifestyle.

## Figures and Tables

**Figure 1 nutrients-13-03408-f001:**
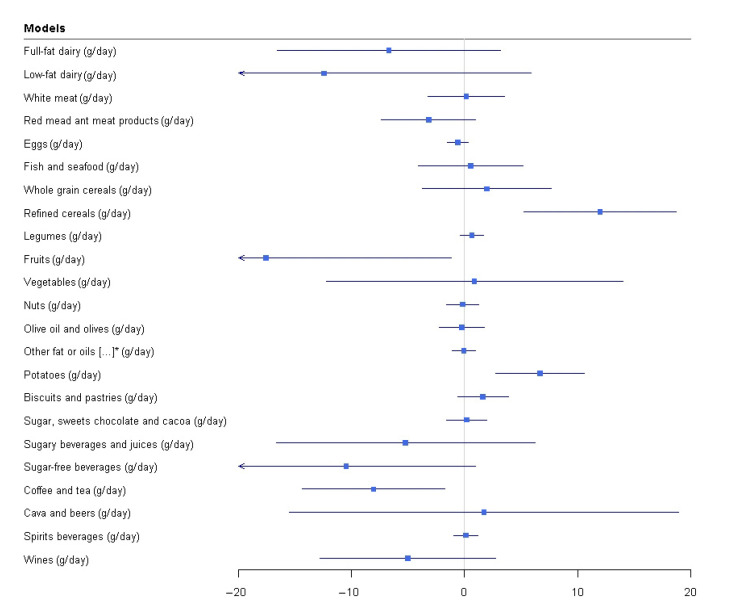
Relationship between CSI and food groups consumption. * other fat or oils, full-fat dairy derivatives and processed meal Multiple linear regressions. CSI (low high deprivation index) as exposure and food consumption (food groups, g/d) as outcome adjusted by age (years), sex (men/women). Somking (smoker, former or never smoked), waist circumference (cm). physical activity (low/high), sedentary lifestyle (no/yes), hypertension(no/yes), dyslipidemia (no/yes), dyslipidemia (no/yes), and type 2 diabetes mellitus.

**Figure 2 nutrients-13-03408-f002:**
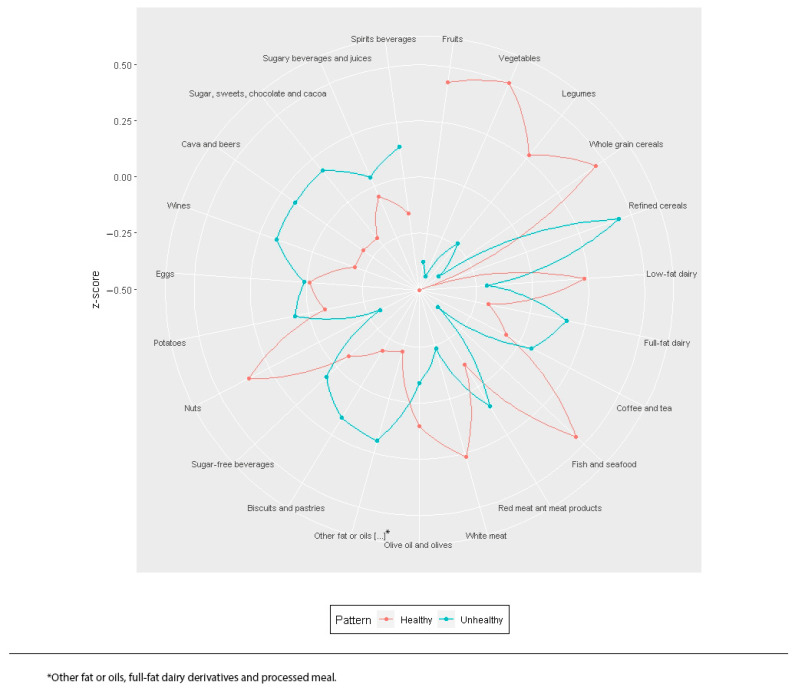
Dietary patterns identified by cluster analysis. * other fat or oils, full-fat dairy derivatives and processed meal.

**Table 1 nutrients-13-03408-t001:** General characteristics of participants according to composed socioeconomic index categories.

	All	High Deprivation Index (≥2.27 Points)	Low Deprivation Index (<2.27 Points)	*p*-Value *^#^*
	*n* = 1512	*n* = 744	*n* = 768	
**Socio-demographic variables**				
Women	759 (50.2%)	378 (50.8%)	381 (49.6%)	0.679
Age (years)	65.5 (4.80)	65.3 (4.83)	65.8 (4.77)	0.041
Civil status *				
Single or religious	65 (4.31%)	32 (4.31%)	33 (4.31%)	0.794
Married	1145 (75.9%)	569 (76.6%)	576 (75.2%)	
Divorced or widowed	299 (19.8%)	142 (19.1%)	157 (20.5%)	
Education level *				
Academic or graduate	345 (22.9%)	113 (15.3%)	232 (30.2%)	<0.001
Secondary education	480 (31.9%)	213 (28.9%)	267 (34.8%)	
Primary education or less	680 (45.2%)	412 (55.8%)	268 (34.9%)	
Employment status *				
Currently working	304 (20.2%)	143 (19.3%)	161 (21.1%)	0.002
Disability	20 (1.33%)	14 (1.89%)	6 (0.79%)	
Housework	147 (9.77%)	92 (12.4%)	55 (7.20%)	
Retired	955 (63.5%)	450 (60.8%)	505 (66.1%)	
Unemployed	78 (5.19%)	41 (5.54%)	37 (4.84%)	
**Anthropometric measurements**				
BMI *				
Mean Kg/m^2^	32.4 [30.1;35.1]	32.4 [30.2;34.9]	32.4 [30.0;35.4]	0.883
≥27 Kg/m^2^	1494 (99.6%)	729 (99.6%)	765 (99.6%)	1.000
Waist circumference *				
Men (cm)	110 [104;116]	110 [104;116]	111 [105;118]	0.136
Women (cm)	104 [98.1;111]	104 [98.0;110]	105 [98.5;112]	0.084
Central obesity	1404 (93.5%)	689 (93.7%)	715 (93.2%)	0.761
**Lifestyle**				
Adherence to the erMedDiet (score from 0 to 17 points)	7.86 (2.51)	7.99 (2.52)	7.74 (2.50)	0.046
Physical activity (METs/Min/week)				
Light	839 [224;1678]	671 [112;1343]	839 [280;1678]	<0.001
Moderate	140 [0.00;1119]	140 [0.00;1171]	43.7 [0.00;1084]	0.196
Vigorous	83.9 [0.00;1119]	72.3 [0.00;934]	112 [0.00;1259]	0.695
Total	2098 [1105;3525]	1979 [1069;3357]	2241 [1133;3776]	0.044
Low physical activity	763 (50.5%)	394 (53.0%)	369 (48.0%)	0.063
High physical activity	749 (49.5%)	350 (47.0%)	399 (52.0%)	
Sedentary lifestyle	675 (44.7%)	299 (40.3%)	376 (49.9%)	0.001
Daily sleeping time (h/day) *	7.00 [6.00;8.00]	7.00 [6.00;8.00]	7.00 [6.00;8.00]	0.583
Smoking *				0.972
Current smoker	171 (11.3%)	84 (11.3%)	87 (11.4%)	
Former smoker	617 (40.9%)	302 (40.6%)	315 (41.1%)	
Never smoked	722 (47.8%)	358 (48.1%)	364 (47.5%)	
**Comorbidities**				
Dyslipidemia *	1042 (69.0%)	520 (70.0%)	522 (68.0%)	0.428
Hypertension	1318 (87.2%)	635 (85.3%)	683 (88.9%)	0.045
Type 2 diabetes mellitus	438 (29.0%)	204 (27.4%)	234 (30.5%)	0.211

Abbreviations: BMI: Body index mass; erMedDiet: energy reduced Mediterranean diet; METs: Metabolic Equivalents. Central obesity: waist circumference men >102 cm and women >88 cm. Data are presented as mean (SD) or median [IR] for continuous variables, and as *n* (%) for categorical variables. * The percentage of missing values was between 0.13% and 0.79% from total study population. *p*-value ^#^: *t*-tests or Mann–Whitney U test for continuous variables; and χ^2^ test and Fisher test for categorical variables.

**Table 2 nutrients-13-03408-t002:** Food consumption of participants according to Composite Socioeconomic Index (CSI) categories.

	All	High Deprivation Index (≥2.27 Points)	Low Deprivation Index (<2.27 Points)	*p*-Value ^#^
	*n* = 1512	*n* = 744	*n* = 768	
Full-fat dairy (g/day)	62.1 [34.6;115]	58.0 [32.6;111]	66.4 [37.0;121]	0.021
Low-fat dairy (g/day)	208 [83.0;319]	204 [68.1;315]	213 [98.3;322]	0.082
White meat (g/day)	68.5 [40.9;84.6]	70.2 [42.2;84.7]	67.6 [39.6;84.2]	0.686
Red meat and meat products (g/day)	88.0 [65.6;120]	85.9 [63.9;115]	92.0 [67.6;123]	0.022
Eggs (g/day)	25.1 [22.0;26.7]	25.0 [21.6;26.6]	25.2 [22.4;26.8]	0.065
Fish and seafood (g/day)	109 [74.8;143]	108 [75.5;143]	109 [73.7;143]	0.963
Whole grain cereals (g/day)	7.39 [0.22;73.9]	5.22 [0.00;67.7]	10.8 [0.38;74.1]	0.046
Refined cereals (g/day)	103 [61.1;155]	108 [63.4;167]	98.3 [59.5;143]	0.004
Legumes (g/day)	17.4 [13.0;24.1]	17.9 [13.3;24.2]	17.0 [12.9;24.1]	0.100
Fruits (g/day)	299 [200;401]	289 [194;377]	311 [209;413]	0.008
Vegetables (g/day)	305 [233;395]	301 [234;396]	310 [231;394]	0.918
Nuts (g/day)	8.87 [3.90;20.4]	9.00 [4.32;20.0]	8.75 [3.58;20.6]	0.378
Olive oil and olives (g/day)	56.5 [45.1;66.8]	55.1 [44.4;66.8]	57.2 [45.9;66.7]	0.175
Other fat or oils, full-fat dairy derivatives and processed meal (g/day)	6.46 [3.13;11.7]	6.49 [3.15;11.9]	6.36 [3.11;11.4]	0.895
Potatoes (g/day)	90.7 [47.2;103]	92.0 [51.8;104]	90.3 [42.9;102]	0.014
Biscuits and pastries (g/day)	14.4 [6.33;27.0]	15.1 [6.77;29.1]	13.8 [6.02;24.3]	0.023
Sugar, sweets, chocolate and cocoa (g/day)	14.4 [6.13;26.9]	14.8 [6.12;27.3]	14.2 [6.14;26.7]	0.831
Sugary beverages and juices (g/day)	35.4 [9.19;112]	36.7 [8.93;104]	34.6 [9.67;115]	0.750
Sugar-free beverages (g/day)	1.29 [0.00;13.3]	1.03 [0.00;11.6]	1.67 [0.00;15.0]	0.002
Coffee and tea (g/day)	92.9 [48.7;127]	71.0 [47.3;126]	100 [50.4;129]	0.005
Cava and beers (g/day)	50.7 [11.9;119]	48.4 [7.45;117]	53.5 [16.5;121]	0.129
Spirits beverages (g/day)	1.05 [0.00;3.10]	0.91 [0.00;3.11]	1.26 [0.00;3.09]	0.040
Wines (g/day)	27.6 [5.65;76.0]	24.4 [1.98;69.5]	30.6 [8.19;78.9]	0.014

Data are presented as median [IR]. *p*-value ^#^: Mann–Whitney U test.

**Table 3 nutrients-13-03408-t003:** General characteristics of the participants according to the lifestyle categories.

	Unhealthy Dietary Pattern and Low Physical Activity	Unhealthy Dietary Pattern and High Physical Activity	Healthy Dietary Pattern and Low Physical Activity	Healthy Dietary Pattern and High Physical Activity	*p*-Overall ^#^
	*n* = 435	*n* = 373	*n* = 328	*n* = 376	
**Socio-demographic variables**					
Women	192 (44.1%)	118 (31.6%) ^a^	224 (68.3%) ^a^	225 (59.8%) ^a,b,c^	<0.001
Age (years)	65.1 (5.08)	65.1 (5.05)	65.8 (4.63)	66.1 (4.27) ^a,b^	0.003
Civil status		^a^	^b^		0.024
Single or religious	20 (4.61%)	13 (3.49%)	15 (4.57%)	17 (4.55%)	
Married	312 (71.9%)	307 (82.3%)	239 (72.9%)	287 (76.7%)	
Divorced or widowed	102 (23.5%)	53 (14.2%)	74 (22.6%)	70 (18.7%)	
Education level *					0.112
Academic or graduate	108 (24.8%)	90 (24.2%)	68 (20.8%)	79 (21.3%)	
Secondary education	149 (34.3%)	124 (33.3%)	91 (27.8%)	116 (31.3%)	
Illiterate or primary education	178 (40.9%)	158 (42.5%)	168 (51.4%)	176 (47.4%)	
Employment status *		^a^	^a^	^a,b,c^	0.001
Currently working	129 (29.8%)	64 (17.3%)	62 (19.0%)	49 (13.1%)	
Disability	7 (1.62%)	4 (1.08%)	9 (2.76%)	0 (0.00%)	
Housework	39 (9.01%)	27 (7.30%)	36 (11.0%)	45 (12.0%)	
Retires	241 (55.7%)	249 (67.3%)	200 (61.3%)	265 (70.7%)	
Unemployed	17 (3.93%)	26 (7.03%)	19 (5.83%)	16 (4.27%)	
**CSI**					0.270
High deprivation index OYWX(≥2.27 points)	227 (52.2%)	177 (47.5%)	167 (50.9%)	173 (46.0%)	
Low deprivation index OYWX(<2.27 points)	208 (47.8%)	196 (52.5%)	161 (49.1%)	203 (54.0%)	
CSI (score)	2.48 [1.83;3.26]	2.26 [1.83;3.22]	2.40 [1.94;3.07]	2.26 [1.78;3.12]	0.436
**Anthropometric measurements**					
BMI *					
Kg/m^2^	33.1 [30.3;35.8]	31.6 [29.6;34.2] ^a^	33.0 [30.5;36.1] ^b^	32.1 [30.0;34.5] ^a,c^	<0.001
≥27 Kg/m^2^	432 (100%)	371 (99.7%)	321 (99.1%)	370 (99.5%)	0.169
Waist circumference *					
Men (cm)	113 [106;119]	109 [104;115] ^a^	111 [106;116]	109 [104;116] ^a^	<0.001
Women (cm)	106 [98.2;114]	104 [97.6;110]	106 [100;111]	103 [97.0;109]	0.053
Central obesity	411 (94.7%)	331 (88.7%) ^a^	311 (96.3%) ^b^	351 (94.4%) ^a,b^	<0.001
**Lifestyle**					
Adherence to erMedDiet (score)	7.00 [5.00;8.00]	7.00 [5.00;8.00]	9.00 [7.00;10.0] ^a,b^	9.00 [8.00;11.0] ^a,b^	<0.001
Sedentary lifestyle	226 (52.0%)	157 (42.1%) ^a^	159 (48.8%) ^b^	133 (35.4%) ^a,b^	<0.001
Daily sleeping time (h/day) *	7.00 [6.00;8.00]	7.00 [6.00;8.00]	7.00 [6.00;8.00]	7.00 [6.00;8.00]	0.743
Smoking *		^a^	^a,b^	^b^	<0.001
Smoker	65 (15.0%)	48 (12.9%)	28 (8.54%)	30 (8.00%)	
Former smoker	172 (39.6%)	182 (48.8%)	108 (32.9%)	155 (41.3%)	
Never smoked	197 (45.4%)	143 (38.3%)	192 (58.5%)	190 (50.7%)	
**Clinical morbidities**					
Dyslipidemia *	285 (65.5%)	248 (66.5%)	235 (71.9%)	274 (72.9%)	0.059
Hypertension	386 (88.7%)	334 (89.5%)	273 (83.2%)	325 (86.4%)	0.056
Type 2 diabetes mellitus	114 (26.2%)	106 (28.4%)	111 (33.8%)	107 (28.5%)	0.139

BMI: Body index mass; erMedDiet: energy reduced Mediterranean diet; METs: Metabolic Equivalents. Central obesity: waist circumference men >102 cm and women >88 cm. Data are presented as mean (SD) or median [IR] for continuous variables, and as *n* (%) for categorical variables. * The percentage of missing values was between 0.13% and 0.79% from total study population. *p*-overall *^#^:* ANOVA or Kruskall–Wallis test for continuous variables, and χ^2^ test or Fisher for categorical variables. For post-hoc comparisons: Tukey or Benjamini and Hochberg. ^a^, significant differences (*p*-value < 0.05) between-groups: *ref.* unhealthy pattern and low physical activity category. ^b^, significant differences (*p*-value < 0.05) between-groups: *ref.* unhealthy dietary pattern and high physical activity. ^c^, significant differences (*p*-value < 0.05) between-groups: *ref.* healthy pattern and low physical activity category.

**Table 4 nutrients-13-03408-t004:** Relationship between CSI categories and lifestyle (dietary patterns + physical activity).

	Healthy Dietary Pattern and High Physical Activity	Unhealthy Dietary Pattern and Low Physical Activity OR [95% CI]	Unhealthy Dietary Pattern and High Physical Activity OR [95% CI]	Healthy Dietary Pattern and Low Physical Activity OR [95% CI]
CSI (high deprivation index)	ref.	1.42 [1.06,1.89] *	1.09 [0.81,1.48]	1.24 [0.91,1.68]
Sex (women)	ref.	0.66 [0.47,0.94] *	0.31 [0.22,0.46] **	1.60 [1.10,2.34] *
Age (years)	ref.	0.97 [0.94,1.00]	0.97 [0.94,1.00]	0.98 [0.94,1.01]
Smoking (former smoker)	ref.	0.47 [0.28,0.79] *	0.67 [0.40,1.14]	0.72 [0.40,1.29]
Smoking (never smoked)	ref.	0.67 [0.39,1.12]	0.79 [0.46,1.37]	0.97 [0.54,1.76]
Waist circumference (cm)	ref.	1.03 [1.03,1.01] **	1.00 [0.98,1.02]	1.01 [1.00,1.03]
Sedentary lifestyle (yes)	ref.	1.81 [1.34,2.43] **	1.20 [0.88,1.63]	1.75 [1.28,2.39]
Hypertension (yes)	ref.	1.33 [0.85,2.06]	1.39 [0.88,2.20]	0.82 [0.53,1.26]
Dyslipidemia (yes)	ref.	0.79 [0.57,1.08]	0.87 [0.63,1.20]	0.93 [0.66,1.31]
Type 2 diabetes mellitus (yes)	ref.	0.77 [0.56,1.07]	0.89 [0.63,1.24]	1.21 [0.86,1.69]

Multinomial regression model. CSI (low/high deprivation index) as exposure and lifestyle (healthy dietary pattern and high physical activity, unhealthy dietary pattern and low physical activity, unhealthy dietary pattern and high physical activity, healthy dietary pattern and low physical activity) as outcome, adjusted by age (years), sex (men/women), smoking (smoker, former or never smoked), waist circumference (cm), sedentary lifestyle (no/yes), hypertension (no/yes), dyslipidemia (no/yes) y type 2 diabetes mellitus (no/yes). Odds ratio (OR) and 95% confidence interval [CI] are shown. ** *p* < 0.001; * *p*-value < 0.05.

## Data Availability

The datasets generated and analyzed during the current study are not expected to be made available outside the core research group, as neither participants’ consent forms or ethics approval included permission for open access. However, the researchers will follow a controlled data sharing collaboration model, as in the informed consent participants agreed with a controlled collaboration with other investigators for research related to the project’s aims. Therefore, investigators who are interested in this study can contact the PREDIMED-Plus Steering Committee by sending a request letter to predimed_plus_scommittee@googlegroups.com. A data sharing agreement indicating the characteristics of the collaboration and data management will be completed for the proposals that are approved by the Steering Committee.

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
