# Peer review of "Deprivation Index and Lifestyle: Baseline Cross-Sectional Analysis of the PREDIMED-Plus Catalonia Study"

_nutrients, 2021, doi:10.3390/nu13103408_

Round 1
Reviewer 1 Report
The authors investigated how unhealthy and healthy diet is related to socio-economical factors and lifestyle. This is highly relevant and deserve attention.
I have a major comment:
- In the introduction large space has been dedicated to differences between genders. In the results Table 4 shows a striking effect of sex, but in the discussion there is no mention to the gender differences. I recommend to spend one paragraph to discuss more these results.
As minor comments:
- In Table 1, civil status numbers do not add up to the total, if there are missing values it is worth mentioning
- Fig 1 is too low quality to read the captions
- page 10, row 268, CIS is mistyped instead of CSI
Author Response
Dear Reviewer,
We sincerely thank to the reviewer the valuable comments and suggestions provided in the following lines, which have helped us to improve our m/s. We addressed all these comments in each of the following points, as well as in the m/s when required.
In adittion, we have had the manuscript systemically revised by a native English provider of editorial assistance. As such, we hope that all errors of grammar, style and have been revomed.
We hope that the revised m/s has reached the quality requirements for inclusion in Nutrients
Yours sincerely,
Corresponding Authors
Responses to the Reviewer 1
The authors investigated how unhealthy and healthy diet is related to socio-economical factors and lifestyle. This is highly relevant and deserve attention.
We sincerely thank to the reviewer the valuable comments and suggestions provided in the following lines, which have helped us to improve our m/s. We addressed all these comments in each of the following points, as well as in the m/s when required.
I have a major comment:
Point 1. In the introduction large space has been dedicated to differences between genders. In the results Table 4 shows a striking effect of sex, but in the discussion there is no mention to the gender differences. I recommend to spend one paragraph to discuss more these results. A paragraph to discuss our findings on the relationship between gender and lifestyle has been included in the Discussion section of the new m/s according to the reviewer comments. (Line 388, page 18)
As minor comments:
Point 2. In Table 1, civil status numbers do not add up to the total, if there are missing values it is worth mentioning. Variables with missing values have been identified, and the percentage of missing values with respect to the total mentioned as footnote in table 1 and 3 of the new m/s.
Point 3. Fig 1 is too low quality to read the captions. Quality of the figures has been improved and included in the new m/s.
Point 4. page 10, row 268, CIS is mistyped instead of CSI. This has been corrected in the new m/s. (Line 335, page 16)
Reviewer 2 Report
Composite Socioeconomic index (CSI) diet and physical activity in Catalonia.
Not clear what is new about this. Seems quite well established that in many populations lower SES is associated with less healthy lfestsyle.
The introduction talks about CSI but nowhere states clearly this is a population based index. It was only evident that this was the case in the detailed description of the CSI in the methods.
How were body weight and height determined? Self-report or measurement?
More information required on the questionnaire for energy-reduced Med Diet. What is this?
At start of statistical analysis it seems data is compared across groups but it is not stated what these groups are.
An analysis with CSI as the outcome and food group intake as the exposure was described but the logic of this is unclear. How could food intake affect CSI? Then you adjust for waist circumference which would not make sense in any direction. How would waist circumference ever be a confounder of CSI implying waist causes or contributes to CSI.
When you look at lifestyle and CSI which is exposure? Are diet and physical activity somehow combined into a single score? It is not clear how this was analysed. It is described in results but needs to be in methods.
Why do you need both the er Med Diet and the cluster dietary patterns?
At the start of the discussion the results are summarised as “high deprivation index was negatively related to the consumption of refined cereals, potatoes, fruits, and coffee and tea” this is not what is indicated in Fig 1 where coffee and tea and fruit are associated with CSI in the opposite direction to potatoes and refined cereal.
In second para of discussion you have described an Australian study where healthy diet was inversely associated with deprivation and then an Australian study where no association between dietary patterns and deprivation. This is not very informative if there is no further information on why they gave opposite results. Otherwise just say findings on association between measures of deprivation and diet patterns have been inconsistent.
Weaknesses talk about assumptions in principal components analysis but this was not used.
The findings of this study can only be applied to the overweight/obese people at increased risk of CVD who participated in the PREDIMED Study. This needs to be acknowledged.
Table 3 is very messy and difficult to read. Should be oriented to landscape and multiple fonts removed to make it easier to look at.
Author Response
Dear Reviewer,
We sincerely thank to the reviewer the valuable comments and suggestions provided in the following lines, which have helped us to improve our m/s. We addressed all these comments in each of the following points, as well as in the m/s when required.
In adittion, we have had the manuscript systemically revised by a native English provider of editorial assistance. As such, we hope that all errors of grammar, style and have been revomed.
We hope that the revised m/s has reached the quality requirements for inclusion in Nutrients
Yours sincerely,
Corresponding Authors
Responses to the Reviewer 2
Point 1. Not clear what is new about this. Seems quite well established that in many populations lower SES is associated with less healthy lfestsyle. The impact in the analysis of the association between the deprivation index with lifestyles has been included in the Introduction section of the new m/s according to the reviewer comments. (Line 97, page 2).
Point 2. The introduction talks about CSI but nowhere states clearly this is a population based index. It was only evident that this was the case in the detailed description of the CSI in the methods. A description of the CSI has been included in the Introduction section of the new m/s according to the reviewer comments (Line 85, page 2)
Point 3. How were body weight and height determined? Self-report or measurement? Body weight, height, and waist circumference were measured by trained staff and following the PREDIMED-Plus operations protocol. We have made the appropriate changes in the Methods Section of the new m/s. (Line 154, page 4)
Point 4. More information required on the questionnaire for energy-reduced Med Diet. What is this? More information about energy-reduced MedDiet (er-MedDiet) questionnaire has been included, as suggested, in the Methods Section of the new m/s. (Line 172, page 4).
Point 5. At start of statistical analysis it seems data is compared across groups but it is not stated what these groups are. Variables of the study were compared across different the groups CSI, food groups and lifestyles categories. This has been clarified in the Statistical analysis section of the new m/s. (Line 207, page 5).
Point 6. An analysis with CSI as the outcome and food group intake as the exposure was described but the logic of this is unclear. How could food intake affect CSI? Then you adjust for waist circumference which would not make sense in any direction. How would waist circumference ever be a confounder of CSI implying waist causes or contributes to CSI. The analysis was made with the CSI as the exposure and food intake as the outcome. We made the appropriate changes in the Statistical analysis section of the new m/s. (Line 228, page 5)
Point 7. When you look at lifestyle and CSI which is exposure? The analysis was conducted with the CSI as the exposure and lifestyle as the outcome. We have made the appropriate changes in the Statistical analysis section of the new m/s. (Line 237, page 5).
Are diet and physical activity somehow combined into a single score? It is not clear how this was analysed. It is described in results but needs to be in methods. We have described how all the groups were created in the new Statistical analysis section, and made some changes to clarify these aspects in the Results section of the new m/s. (Line 229, page 5)
Point 8. Why do you need both the er Med Diet and the cluster dietary patterns? In our analysis, we wanted to evaluate an “a priori” dietary pattern (er-MeDiet) and “a posteriori” dietary pattern based on the participants' food consumption, both offer us complementary information. A posteriori dietary patterns (using statistical test: clusters, etc) offer an alternative approach and a greater relationship with different food groups can be shown, considering the whole diet and how foods are eaten in combination. It should be noted that these statistical methods serve to identify dietary patterns in a particular population, however, the patterns identified do not necessarily represent ideal diet. In contrast, dietary pattern obtained a priori, which includes the use of a score on the quality of the diet is based on dietary recommendations or guidelines. For example, one can cite, among others, Healthy Eating Index, Mediterranean Diet, etc., which assesses the quality by means of an index of the entire diet and not just the components in isolation.
Point 9. At the start of the discussion the results are summarised as “high deprivation index was negatively related to the consumption of refined cereals, potatoes, fruits, and coffee and tea” this is not what is indicated in Fig 1 where coffee and tea and fruit are associated with CSI in the opposite direction to potatoes and refined cereal. We do agree with the reviewer observation. This has been corrected in the Discussion section of the new m/s. (Line 340, page 18).
Point 10. In second para of discussion, you have described an Australian study where healthy diet was inversely associated with deprivation and then an Australian study where no association between dietary patterns and deprivation. This is not very informative if there is no further information on why they gave opposite results. Otherwise just say findings on association between measures of deprivation and diet patterns have been inconsistent. We have modified the paragraph in the Discussion section according to the reviewer comments. (Line 351, page 18)
Point 11. Weaknesses talk about assumptions in principal components analysis, but this was not used. We used cluster analysis as we mentioned in the Statistical analysis section. This has been corrected in the new m/s. Line 409, Page 18)
Point 11. The findings of this study can only be applied to the overweight/obese people at increased risk of CVD who participated in the PREDIMED Study. This needs to be acknowledged. We have included this aspect as a limitation in the Discussion section of the new m/s. (Line 407, Page 19).
Point 12. Table 3 is very messy and difficult to read. Should be oriented to landscape and multiple fonts removed to make it easier to look at. We have modified the Table 3 according to the reviewer comments.
Round 2
Reviewer 2 Report
The introduction still talks about differences in diet and physical activity by age, sex and SES, rather than focussing on the topic of the manuscript which is SES.
Poor diet and physical activity impact on a wider range of health outcomes than just obesity so the introduction should start with this point then that SES was associated with health in other studies, then explain that this could be attributable to some extent to association between SES and lifestyle, which is all well known. Then what is this study going to add to what is already known? You have done this to some extent but given we already know that low SES is associated with poorer health and less healthy lifestyles in general, and that your study sample is quite selected so cannot apply findings to the general population, the specific requirement for this analysis is not well established.
You use two different methods to assess diet which you justified in your response but in the abstract the erMedDiet is not mentioned.
The new section explaining the erMedDiet needs editing. A questionnaire is not ‘determined’ and a questionnaire cannot increase adherence to the diet.
Which variable is the exposure (CSI) and which is the outcome (diet) is now clear but the variables adjusted for seem inappropriate as confounders because many of them cannot be considered common causes of CSI and diet. Why were they included?
The new section describing the creation of 4 categories based on healthy/unhealthy diet and low/hi physical activity needs to be carefully edited. To improve clarity and reduce repetition.
Another problem with this study is apparent in Table 1 where it is shown that there is no difference in measures of obesity between low/hi CSI groups which is the premise of the study but because of the section criteria where all participants are overweight/obese the study cannot properly investigate how CSI relates to lifestyle which might explain differences in overweight and obesity.
Is increasing CSI possible? Is it a realistic intervention?
Author Response
Dear Reviewer
We have read your comments and considered all your suggestions and have made the appropriate changes in the revised version of the manuscript (m/s).
We hope that the revised m/s has reached the quality requirements for inclusion in Nutrients.
Yours sincerely,
Corresponding authors
Point 1. The introduction still talks about differences in diet and physical activity by age, sex and SES, rather than focussing on the topic of the manuscript which is SES. We agree with your observation, and we have modified the Introduction section focusing on SES in the new m/s according to the reviewer comments. However, we decided to mention the differences between diet and physical activity by age, sex, as other social determinants, to address the comment from the reviewer 1.
Point 2. Poor diet and physical activity impact on a wider range of health outcomes than just obesity so the introduction should start with this point then that SES was associated with health in other studies, then explain that this could be attributable to some extent to association between SES and lifestyle, which is all well known. Then what is this study going to add to what is already known? You have done this to some extent but given we already know that low SES is associated with poorer health and less healthy lifestyles in general, and that your study sample is quite selected so cannot apply findings to the general population, the specific requirement for this analysis is not well established. We have modified the Introduction section focusing on SES in the new m/s according to the reviewer points and comments.
In studies of health inequalities, SES is most commonly operationalized as either education, social class, or income, and often without providing a rationale for the choice of indicator. Education, social class, or income can have overlapping properties in relation to health. The deprivation indices are instruments used to measure health inequalities at population level. All of them are constructed based on different socioeconomic or demographic characteristics and are used to quantify the socioeconomic variation in health outcomes. The CSI is a deprivation index constructed by income, occupations, education, and other socioeconomic indicators. Demonstrating that the CSI is associated with patterns of unhealthy lifestyles, diet and physical activity, is important in order to use it as an instrument to prioritize interventions at community level, especially in primary healthcare areas, and it would allow us to have a broad vision of how the socioeconomic contextual aspects of geographic location impact on health related to diet and physical activity.
Point 3. You use two different methods to assess diet which you justified in your response but in the abstract the erMedDiet is not mentioned. We have clarified that two a posteriori dietary patterns were identified by cluster analysis in the Abstract section on the new m/s. The erMedDiet was used only as a descriptive score but not to categorize individuals according to the lifestyle. (Line 37, page 1)
Point 4. The new section explaining erMedDiet needs editing. A questionnaire is not ‘determined’ and a questionnaire cannot increase adherence to the diet. We have made the appropriate changes in the Methods Section of the new m/s. (Line 196, page 4)
Point 5. Which variable is the exposure (CSI) and which is the outcome (diet) is now clear but the variables adjusted for seem inappropriate as confounders because many of them cannot be considered common causes of CSI and diet. Why were they included? We have adjusted by these potential confounders because all are potential determinants of food consumption. This has been mentioned in the Statistical analysis section of the new m/s. (Line 267, page 6).
Point 6. The new section describing the creation of 4 categories based on healthy/unhealthy diet and low/hi physical activity needs to be carefully edited. To improve clarity and reduce repetition. We have made the appropriate changes in the Methods Section of the new m/s. (Line 254, page 5).
Point 7. Another problem with this study is apparent in Table 1 where it is shown that there is no difference in measures of obesity between low/hi CSI groups which is the premise of the study but because of the section criteria where all participants are overweight/obese the study cannot properly investigate how CSI relates to lifestyle which might explain differences in overweight and obesity. We thank the reviewer for their comment. One of the inclusion criteria of PREDIMED-Plus is to have overweight or obesity. Therefore, our study results can only be extrapolated to this type of population. This has been acknowledged as a limitation. In our study, we only try to analyze the associations between CSI and lifestyle, but not to the disease.
Point 8. Is increasing CSI possible? Is it a realistic intervention? All primary healthcare areas registered in Catalonia (n=398) have assigned a CSI. All primary healthcare areas registered in Catalonia (n=398) have assigned a CSI. The CSI ranges from −0.01 to 5.68, a higher value of the CSI implies higher deprivation index. It can be updated more frequently than indices constructed from census variables and is related to the need of health service use.
We cannot change the CSI, but we can adjust lifestyle interventions considering the CSI and prioritize resources. Now, in Catalonia the figure of the nutritionist is prioritized in primary care centers. We are in the debate of what type of health-related interventions should be adjusted in the territorial context. Certainly, we believe that this study provided evidence to implement the PREDIMED-PLUS intervention and adjust it to the context of the territory.